# Psychometric Properties of the Child Neglect Scale and Risk Factors for Child Neglect in Chinese Young Males Who Were Incarcerated

**DOI:** 10.3390/ijerph20054659

**Published:** 2023-03-06

**Authors:** Jinliang Qin, Xi Wang, Chen Chen

**Affiliations:** 1Hangzhou College of Childhood Teachers’ Education, Zhejiang Normal University, Hangzhou 311231, China; 2School of Education Science, Nanjing Normal University, Nanjing 210097, China; 3Institute of Advanced Studies in Humanities and Social Sciences, Beijing Normal University, Zhuhai 519087, China

**Keywords:** juvenile delinquents, child neglect, psychometric testing, reliability, validity, risk factors

## Abstract

Child neglect is an important risk factor for juvenile delinquency, while few studies have examined child neglect in Chinese juvenile delinquents due to the lack of appropriate measurement tools. The Child Neglect Scale is a 38-item retrospective self-report scale that specifically focuses on child neglect. The current study, therefore, aimed to examine the psychometric properties of the Child Neglect Scale and risk factors for child neglect among Chinese juvenile delinquents. A total of 212 young males who were incarcerated participated in this study, and the Childhood Trauma Questionnaire, Child Neglect Scale, and basic information questionnaire were used to collect data. The results showed that the Child Neglect Scale has good reliability, and the mean inter-item correlation coefficients reach accepted standards. Moreover, it is found that child neglect is prevalent among Chinese young males who are incarcerated, with communication neglect occurring most frequently. Low levels of family monthly income and rural residency are risk factors for child neglect. The average scores of security neglect, physical neglect, and communication neglect of the participants respectively show statistically significant differences according to the type of major caregivers. Findings suggest that the Child Neglect Scale may be used to measure child neglect with four independent subscales in Chinese young males who are incarcerated.

## 1. Introduction

Juvenile delinquency is a serious worldwide social issue, which detrimentally influences individual development and social stability. Although most countries have implemented various policies and acts to prevent juvenile delinquency, the occurrence of juvenile delinquency is still high. Each year, 200,000 homicides occur among youths aged 10–29, accounting for 43% of the total annual homicides in the world [1]. Statistics show that the number of juvenile offenders in China in 2020 is 34,000, accounting for 2.21% of the total number of criminals in the same period [2]. Existing studies have suggested that plenty of factors increase the likelihood of juvenile delinquency, especially adverse childhood experiences [3,4,5].

Child neglect, as a type of adverse childhood experience, has the same or even more deleterious consequences than child abuse [6,7]. Previous studies have often thought of neglect and abuse as if they go together or regarded neglect as a category of abuse, but nowadays, more and more researchers have pointed out that neglect is a separate entity, which should be distinguished from abuse and paid significant attention to [8,9,10]. There are many differences between neglect and abuse. For example, neglect, defined as “neglectful failure to supply the needs of children” (an act of omission of caregivers), is unique from abuse (an act of commission of caregivers) [11]. In addition, the consequences of abuse often show signs of physical and mental injuries, which are generally easy to investigate, collect evidence on, and diagnose [12,13], while the consequences of neglect are relatively hidden [8,9]. Therefore, more research focusing on child neglect is necessary to raise our awareness and concern about child neglect.

Although child neglect has been considered a risk factor for juvenile delinquency [14,15], the relationships between specific types of child neglect are inconsistent [16,17,18,19]. Moreover, there are few measurement tools focused especially on child neglect [20]. Additionally, the risk factors of child neglect among youth who are incarcerated need further exploration. Therefore, the current study attempts to validate a child neglect measure among Chinese youths who were incarcerated and delineate risk factors of child neglect among this sample.

### 1.1. Measurements of Child Neglect in the Chinese Cultural Context

Prospective informant reports and retrospective self-reports are important methods to measure child neglect [21]. Some researchers argue that the retrospective self-reports of adverse childhood experiences may be unreliable [21,22,23,24] due to the memory biases caused by forgetting, infantile amnesia, and other reasons [20,25], which may cause them to overestimate or underestimate their adverse childhood experiences [23,26]. Similarly, prospective measures of neglect mainly come from official records, which are not easy to obtain and capture only a small proportion of cases, which may underestimate the prevalence of adverse childhood experiences [24]. Therefore, retrospective self-reports, especially through questionnaires, may be widely used to explore childhood experience.

Several retrospective questionnaires have been used to assess child neglect in a Chinese cultural context. For example, the Childhood Trauma Questionnaire (CTQ), built by Bernstein and colleagues [27], has different subscales to assess child neglect, including physical neglect, and emotional neglect. The CTQ is a good psychometric instrument for assessing child neglect in the sample of Chinese adolescents and adults [28,29], as well as those who are incarcerated [30,31]. However, in the CTQ, there are only 10 items to measure neglect, and the neglect subscale is only divided into two categories: emotional neglect and physical neglect. Hence, the neglect subscale in the CTQ may not be able to give a whole picture of child neglect, which may influence the understanding of child neglect in China.

Additionally, some retrospective questionnaires based on the Chinese cultural context have been developed to measure child neglect. For example, the Child Neglect Scale (CNS) is a measure only focuses on neglect. It was compiled by Yang [10] and consists of four subscales: security neglect, physical neglect, communication neglect, and emotional neglect. Security neglect refers to the neglect of potential safety hazards in children’s growth and living environments; physical neglect refers to the neglect of children’s physical care, such as clothing, food, shelter, etc., and the omission or delay of children’s medical and health care needs; communication neglect refers to the neglect of communication with children; and emotional neglect refers to the neglect of children’s feelings and the lack of satisfaction with children’s emotional needs [10]. The CNS has been used in some Chinese studies to assess child neglect, and it has good reliability and validity in the studies [32,33,34]. However, few studies have used it to assess child neglect among juvenile delinquents in China. Therefore, the current study attempts to apply the CNS to the sample of Chinese youth who were in incarcerated and validate the CNS. We hypothesize that the CNS has good reliability and validity in Chinese young males who were incarcerated. 

### 1.2. The Risk Factors of Child Neglect

A growing body of studies has explored risk factors for child neglect. For example, poverty is a powerful predictor of child neglect, because it means a lack of financial resources, which may limit the ability of the caregivers to provide adequate care and supervision [35,36]. Family income is a crucial index that can reflect the family’s financial situation. Individuals with low levels of family income are more vulnerable to be neglected [37,38]. What is more, compared with non-rural areas, the economic conditions in rural areas are often worse. The prevalence of child neglect is high in rural area of China, and people from rural areas are more likely to experience child neglect than those from non-rural areas [38,39]. 

Moreover, family structure may be a risk factor for child neglect. Some studies suggested that child neglect was more likely to happen in a large family [40,41], while others indicated that the larger number of family members in a household would reduce the risk of child neglect [42]. Li et al. [43] examined family structure as a risk factor for child neglect and found that the highest prevalence of child neglect was observed in children from step families (80.37%), followed by one-parent families (66.41%), nuclear families (56.32%), and extended families (51.58%). Vanderminden et al. [44] pointed out that families with two biological parents had lower child neglect rates than other household configurations. A large number of studies have found that left-behind children report a higher level of child neglect, and most of them are raised by grandparents [43,45].

Similarly, youth who were incarcerated may come from disadvantaged families. For example, Chang et al. [46] based on in-depth interviews with 18 delinquents in custody found that most of the juvenile delinquents were from nontraditional families, such as single-parent families and reconstituted families. Bobba et al. [47] found that 17.71% of juvenile delinquents’ parents were both dead, and 17.71% of juvenile delinquents had only one parent. In addition, left-behind children and migrant children are high-risk groups easily involved in juvenile delinquency. Parental absence (e.g., parental divorce or parental death) may result in more caretaker–child conflicts, negative discipline, and poor supervision [44,48,49].

Therefore, the current study attempts to delineate the risk factors for child neglect among Chinese youths who are incarcerated. We hypothesize that the economic status and family structures may be risk factors for child neglect among Chinese youths who are incarcerated. 

### 1.3. The Current Study

Juvenile delinquency is an important issue for society development, and decreasing juvenile delinquency is an urgent need for any country. China is a developing country, and it is on the way to realizing the Chinese dream, which requires attention to the prevention of juvenile delinquency. Child neglect may be a reason for juvenile delinquency, while few measures could be used to assess child neglect among Chinese youths who are incarcerated. Therefore, the current study aims at conducting psychometric testing with the Child Neglect Scale in Chinese young males who are incarcerated and delineate risk factors for child neglect in a Chinese cultural context.

## 2. Materials and Methods

### 2.1. Participants

A total of 212 youth males who were incarcerated were recruited from the juvenile detention center in Hangzhou, Zhejiang Province, China. Initially, 218 youth males attended the current study, while 6 of those were deleted since they did not complete it or their answers were invalid. The mean age of the sample was 17.49 (SD = 1.52) years old, ranging from 15 to 25 years old. Specifically, 35.8% (76/212) of those were the only child in the family. 87.3% (185/212) of them spent their childhood in the countryside or small cities, and others (10.4%) spent their childhood in the suburbs or big cities. Most of the participants (73.9%, 139/212) reported that their education levels were junior high school or below. Moreover, 30.2% (64/212) of the participants have bad hobbies such as drinking and playing cards/mahjong. Concerning family incomes, 46.2% of participants from families with monthly incomes less than 5000 RMB (USD 704.99), 30.2% from families with monthly incomes ranging from 5001 to 10,000 RMB (USD 705.131 to 1409.98), and the remaining youths (21.7%) from families with monthly incomes greater than 10,000 RMB (USD 1409.98). With regards to the marital status of parents, in the majority of the families, the parents’ marital status was original (both husband and wife were registered for marriage for the first time, 69.3%), while 29.7% of them were from a single-parent family or their parents were remarried. More detailed information on the participants can be found in Table 1.

### 2.2. Procedures

First, the researchers expressed the study’s purposes and presented questionnaires to the leaders and prison guards of Hangzhou Juvenile Detention Center and obtained their permission to conduct the current study after discussing some details. Second, participants were randomly recruited from two districts of the juvenile detention center. Specifically, the authors randomly chose room numbers, and prison guards asked the participants who lived in these rooms to finish the questionnaires. The participants and prison guards who assisted in conducting the current study acknowledged the research aims and procedures, and informed consent was obtained from participants before data collection. Thirdly, participants completed the questionnaires in learning rooms within 30 min. Additional materials such as pencils were provided by the juvenile detention center. During the data collection process, some problems were solved. For example, some juvenile delinquents with low educational levels could not understand the items entirely, so the researchers read and explained the questions to them. The current study was approved by the ethics committee of the researchers’ institution and the juvenile detention center, and the questionnaires and procedures of this study were safe for participants.

### 2.3. Measures

*Child Neglect Scale (CNS).* The CNS is a 38-item retrospective self-report scale used to assess childhood neglect [10]. It has four subtypes to evaluate different kinds of neglect, including emotional neglect (e.g., “*My parents were indifferent to me.*”), physical neglect (e.g., “*My parents took me to see a doctor in time when I was ill.*”), security neglect (e.g., “*Told me something about safety.*”), and communication neglect (e.g., “*When I was punished, I was never given the reason.*”). Participants gave responses to each item based on a 5-point Likert scale: 1 means that the situation reflected by the item never happened, and 5 means that the situation reflected by the item always happens. There are 12 reverse scoring items in the CNS (see Table 2). After reverse scoring, the higher the total score of the CNS and its subscales, the more serious the child neglect or a specific neglect type is. The CNS has good reliability and validity in previous studies based on general samples [50].

*Childhood Trauma Questionnaire-Short Form (CTQ-SF).* The CTQ-SF is a frequently used 28-item self-report instrument used to assess abuse and neglect in childhood. It was compiled by Bernstein and his colleagues [27] and was translated into Chinese by Zhao et al. [51]. The subscales of neglect are divided into two dimensions, which include physical neglect (e.g., “*worn dirty clothes.*”) and emotion neglect (e.g., “*felt loved.*”). Items are rated on a 5-point Likert-type scale (1 = “*never*” to 5 = “*almost always*”), with higher scores for each scale indicating the item reflection happened more frequently. In the current study, the Cronbach’s alpha for the total neglect subscales was good (α = 0.878), and it has good validity (χ^2^/df = 2.139, CFI = 0.958, GFI = 0.934, IFI = 0.958, TLI = 0.942, and RMSEA = 0.073).

*Demographic Information Questionnaire (DIQ).* The DIQ was developed by the authors and used to collect demographic information about the participants, such as height, weight, family income, parents’ marital status, and educational levels.

### 2.4. Data Analysis

Before the data analysis, missing values and outliers were examined, and the questionnaires with missing data (>15%) were excluded [52]. Data analysis was conducted using SPSS 23.0 and AMOS 17.0 software. The Cronbach’s α and mean inter-item correlations (MIC) were used to evaluate the internal consistency of the CNS. In general, Cronbach’s α coefficients above 0.70 are considered acceptable [53]. An optimal range of 0.10–0.40 was set for the MIC [54,55]. The item-total correlations and the correlations between the total scale and the four subscales were also examined using Pearson’s *r*, all tests were two-tailed for significance, and the significance (*p*-value) was set at 0.05.

What is more, a confirmatory factor analysis (CFA) was used to test the factor structure of the CNS. The goodness of model fit was assessed by the Root Mean Square Error of Approximation (RMSEA), and the fit indexes were the Comparative Fit Index (CFI), Goodness of Fit Index (GFI), Incremental Fit Index (IFI), and Tucker Lewis Index (TLI). CFI, TLI, GFI, and IFI values of 0.90 or higher indicated a good fit. RMSEA values of less than 0.08 were considered a good fit [56]. 

In addition, this study used criterion-related validity to ensure the CNS is a good measure for child neglect in youths who were incarcerated, and a descriptive analysis was used to delineate the characteristics of child neglect among Chinese young males who were incarcerated.

## 3. Results

### 3.1. Internal Consistency

The Cronbach’s alpha of the total CNS was 0.939 in the current study, and the internal consistencies for the subscales of security neglect, emotional neglect, physical neglect, and communication neglect were 0.813, 0.897, 0.772, and 0.753, respectively. Meanwhile, the split-half coefficient for the total CNS was 0.941. Moreover, the MIC coefficient for the total CNS was 0.294, and the MIC coefficients for the four subscales ranged from 0.310 (communication neglect) to 0.370 (emotional neglect) (see Table 3).

### 3.2. Correlations among Total CNS and Its Four Facets

All of the item-total correlation coefficients between the total score of each subscale and each item in the subscale were significant (*p* < 0.01), including emotional neglect (0.403 ≤ *r* ≤ 0.755), physical neglect (0.460 ≤ *r* ≤ 0.792), security neglect (0.510 ≤ *r* ≤ 0.746), and communication neglect (0.551 ≤ *r* ≤ 0.775). The correlations among the total CNS and its four facets were presented in Table 4. As indicated, the correlation coefficients of the total score of CNS and its four facets ranged from 0.474 to 0.926 (*p* < 0.01).

### 3.3. Construct Validity

The results of the CFA are presented in Figure 1, and the indexes of the model fit are presented in Table 5. In the four-factor model, the RMSEA = 0.083, CFI = 0.746, IFI = 0.749, GFI = 0.698, and TLI = 0.727. The four-factor model’s fit indices were undesirable. All factor loadings were significant, ranging from 0.287 to 0.773.

Moreover, we also conducted an exploratory factor analysis and tried different models, such as a two-factor model, three-factor model, and second-order model, but none of them had good construct validity.

Additionally, we conducted the CFA for these four independent subscales, and the model fit indices of security neglect, emotional neglect, physical neglect, and communication neglect were acceptable (see Table 5).

### 3.4. Criterion-Related Validity

The results of the Pearson correlation of the CNS and CTQ showed that the total score of the CTQ neglect subscales (21.15 ± 7.96) was significantly correlated with the total score of the CNS (79.35 ± 23.74, *r* = 0.735, *p* < 0.001).

### 3.5. The Risk Factors for Child Neglect

The means and standard deviations of the child neglect scores of all the participants are shown in Table 6. The average score on the CNS ranged from 1.00 to 3.89 (Mean = 2.09; SD = 0.63). Whereas communication neglect had the highest mean score, physical neglect had the lowest mean score.

Moreover, the juvenile delinquents who lived in rural areas during childhood scored significantly higher on the average score of the overall CNS than those who lived in non-rural areas, *t*_(205)_ = −2.34, *p* = 0.02 < 0.05, *d* = −0.33. Except for emotional neglect (*t*_(205)_ = −1.68, *p* > 0.05), the juvenile delinquents living in rural areas scored significantly higher on the other subscales of the CNS. However, the results indicated that there were no significant differences in the score of the overall CNS or a specific subscale in relation to whether they are an only child or not (*p* > 0.05); see Table 6. 

Table 7 shows the one-way ANOVA results of child neglect among juvenile delinquents. No significant difference was found in the average scores of the overall CNS and its subscales of the participants in the study according to age. However, the average scores of total neglect (*F* = 3.66, *p* = 0.027 < 0.05, *η*^2^ = 0.03), security neglect (*F* = 5.07, *p* = 0.007 < 0.01, *η*^2^ = 0.05), and physical neglect (*F* = 8.30, *p* = 0.000 < 0.01, *η*^2^ = 0.07) of the participants showed a statistically significant difference according to educational levels. Moreover, the levels of security neglect, physical neglect, and total neglect of those with educational levels below junior high school were significantly higher than those with educational levels at and above junior high school. In addition, a main effect emerged for the group, *F* (2, 207) = 3.34, *p* < 0.05, *η*^2^ = 0.05, in which juvenile delinquents with monthly household incomes of 3001–5000 RMB (USD 423.135–704.99) and 5001–10,000 RMB (USD 705.131–1409.98) had significantly higher scores on the CNS than those with a monthly household incomes of more than 10,000 RMB (USD 1409.98), and juvenile delinquents with monthly household incomes of 3001–5000 RMB also had significantly higher scores on the emotional neglect subscale than those with a monthly household income of more than 10,000 RMB.

Moreover, the average scores of security neglect (*F* = 5.36, *p* = 0.000 < 0.001, *η*^2^ = 0.11), physical neglect (*F* = 6.67, *p* = 0.000 < 0.001, *η*^2^ = 0.12), and communication neglect (*F* = 5.07, *p* = 0.000 < 0.001, *η*^2^ = 0.10) of the participants, respectively, showed a statistically significant difference according to the type of major caregivers. In terms of security neglect, the scores of participants whose major caregivers are parents and grandparents were significantly lower than those of participants whose major caregivers are mother, grandparents, or others, and participants whose grandparents are the major caregivers scored significantly higher than those whose major caregivers are parents. In terms of physical neglect, participants whose major caregivers are parents and grandparents scored significantly lower than those whose major caregivers were all other types, and participants whose grandparents are the major caregivers scored significantly higher than those whose major caregivers are parents. Concerning communication neglect, the scores of participants whose major caregivers are parents and grandparents were significantly lower than those of participants whose major caregivers are their mother or grandparents, and participants whose parents are the major caregivers scored significantly lower than those whose major caregivers are their mother or grandparents.

## 4. Discussion

The current study examined the psychological properties of the CNS and delineated the risk factors for child neglect among Chinese juvenile delinquents. The results showed that the CNS has good reliability but poor validity in a four-factor model among Chinese young males who were incarcerated. Therefore, the CNS may have four independent subscales for measuring child neglect in juvenile delinquents in a Chinese cultural context. 

The results of the reliability showed that the Cronbach’s α coefficients for each subscale and total scale of the CNS reached accepted standards (α > 0.70), and all MIC coefficients were between 0.294 and 0.370, which indicates that the CNS has good internal consistency in young males who are incarcerated [53,54,55]. Moreover, the correlation coefficients between subscales and the total scale were between 0.474 and 0.926, which reflects the high reliability of the CNS in Chinese juvenile delinquents. These results are consistent with previous studies conducted in general samples [50,57]. 

The correlation coefficients of the CNS and CTQ were significantly high (r = 0.735, *p* < 0.001), which suggests that the CNS has ideal Criterion-related validity. However, the CFA results indicated that the four-factor model was not confirmed in Chinese young males who were incarcerated. Although each subscale of the CNS has desirable model fit indices, the four-factor model did not fit the data well. The unsatisfactory CFA results of the four-factor model mean that the overall score of the CNS may not accurately and truly show the characteristics of child neglect of juvenile delinquents in this study, but the neglect reflected by each subscale is relatively reliable.

The reason for the poor fit between the four-factor model and the data may be the dependence of some fit indices on the sample size. In previous studies using the CNS, the sample size was generally about 500 to 1000 subjects [10,50,57]. In this study, the sample size only consisted of 212 subjects, which may lead to undesirable model fit indices. Another reason may be the model structure is not suitable. It can be found that there is a high correlation between physical neglect and security neglect, and between emotional neglect and communication neglect in this study. Reviewing the process of compiling the CNS, Yang [10] did not include all the items of the whole scale in factor analysis at one time but carried out the factor analysis in different dimensions with the items of each subscale, which may give rise to the appropriate model structure. Finally, it cannot be ignored that the education level of participants is low. They may not understand or misunderstand the meaning of the item because of the expression of the item. For example, the N35 “When I told my parents that I had been bullied by my peers, my parents ignored me” belongs to the security neglect subscale, but the expression is very similar to the item in communication neglect subscale (e.g., N33 “My parents ignored me when I asked questions I didn’t understand.”). Therefore, the construct validity of CNS may need further verification in a larger sample of juvenile delinquents with higher educational levels. The structure of the child neglect model (including which types of neglect are covered and the relationship between different types of neglect) should be adjusted and the items of each subscale of the CNS should be revised and appropriately classified.

The majority of participants reported that they have experienced one or more kinds of child neglect. As assumed, the results show that child neglect scores are much higher in our sample (SN: 2.11; PN: 1.78; EN: 2.17; CN: 2.20) than in a non-delinquent sample (SN: 1.55; PN: 1.48; EN: 1.65; CN: 1.76) [58]. In line with the previous research in a general sample, the mean score of communication neglect was higher than other types of neglect [32,34]. These findings suggest that child neglect, especially communication neglect is pervasive among Chinese young males who were incarcerated. Thus, caregivers should pay close attention to parent-child communication and emotional warmth, rather than just ensuring material guarantee.

In addition, the results of this study supported the hypothesis that lower family income and rural residency are both risk factors for child neglect. A monthly household income of less than 5000 RMB means that the family economic level is in poverty, and a monthly household income ranging from 5001 to 10,000 RMB means that the family’s economic level is comparatively well-off, which is the level of most families in China. Participants from poor and comparatively well-off families are more likely to be neglected during their childhood [38]. Low levels of family income are a barrier that prevents parents from providing adequate supervision and care to children. Since, under such circumstances, parents need to concentrate on their work to gain financial resources, they may not be able to take good care of their children’s daily lives and meet their emotional needs. Moreover, participants who lived in rural areas during childhood were easier to be neglected than those who lived in non-rural areas, which was consistent with prior studies [59]. On the one hand, the economic level in rural areas is relatively poor; on the other hand, a considerable number of children in rural areas are left-behind children. Their parents have moved to the cities for a living so the possibility of parents’ supervision and concern for them being reduced is greatly increased. It invisibly increased the likelihood of child neglect. 

Meanwhile, this study suggested juvenile delinquents whose major caregivers are parents and grandparents or parents may experience milder child neglect than those major caregivers are other types, especially grandparents. Different types of major caregivers are the important features that distinguish different family-rearing patterns. In the mode of “co-parenting”, the major caregivers of children are parents and grandparents, which are more common in stem families. In the mode of “parental parenting”, the major caregivers of children are parents, which are more common in nuclear families. In the mode of “grandparenting”, the major caregivers of children are grandparents, which are more common in families of left-behind children. Child neglect is easier to occur in the mode of “grandparenting”, which is affected by multiple factors. First of all, the grandparents’ generation has a relatively low education level, and their concept of family upbringing is backward, so there are often some unscientific upbringing behaviors, such as only paying attention to children’s basic needs such as “eating and wearing warm clothes”. Second, after all, the love given by grandparents cannot replace the love of parents. Moreover, limited by their physical condition and energy, the grandparents’ supervision and care of children may be inadequate. The reason why the “co-parenting” mode is better than the other two modes of family parenting is that the cooperation and participation of family members can create a harmonious family atmosphere, and family members can urge each other in their parenting behaviors, thus effectively preventing child neglect and foster children’s development. 

Considering the prevalence of child neglect among juvenile delinquents and various potential factors affecting child neglect, the caregivers and the national government should take some measures toward the prevention and intervention of child neglect. Currently, the measures commonly used in the world to prevent child neglect are the home visiting evidence-based program (i.e., the SafeCare in Spain) [60], and establishing specialized agencies and organizations. While enacting national laws [61], our country can also adopt various prevention and intervention programs such as home visiting programs. In addition, we should strengthen the protection of the rights of disadvantaged children, such as left-behind children, children from divorced families, and so on. Parents or parents and grandparents should be encouraged to raise children together, and accept scientific parenting guidance for families.

However, it is different from the previous research [43,62], whether the only child or not is not associated with the occurrence of child neglect in the current study. There are several possible reasons for this result. First, with the accelerated pace of life, the contradiction between parents’ work and children’s care has become increasingly prominent [63]. Whether raising one child or multiple children, parents do not have enough time to take care of and accompany their children. Second, being influenced by the Chinese traditional culture (Dragon Sons, Phoenix Daughters), Chinese parents often place high expectations on their children, hoping that they will excel in their studies and stand out in the future. Therefore, they may neglect emotional support and communication with children while meeting the needs for material and education. This phenomenon exists both in one-child and multi-child families.

Although the current study may be the first study to evaluate the psychometric properties of the CNS in Chinese juvenile delinquents, some limitations should be considered. First, the participants of the current study just included males who are incarcerated, which may not present the whole picture of the psychometric properties of the CNS in juvenile delinquents. Evidence shows that the relationship between adverse childhood experiences and delinquent behavior may be different between males and females [64,65]. Future studies should expand the sample population and add females into the studies, and validate the measures among both males and females who were incarcerated. Second, the current study did not divide the participants into different groups based on their criminal types, which may not present a clear relation between overall child neglect and criminal types and even more, the relationship between specific neglect types and crime types. Future studies are needed to divide juvenile delinquents into several groups based on their criminal types, and explore relations between subtypes of child neglect and criminal types. Finally, in the future, the retrospective study of child neglect using the CNS can be combined with interviews to improve the reliability of the results. 

## 5. Conclusions

In conclusion, the current study shows that the CNS has good reliability with an undesirable validity of a four-factor model but may be used for child neglect with independent subscales in Chinese young males who are incarcerated. Moreover, child neglect is pervasive among Chinese young males who are incarcerated, with communication neglect occurring most frequently. Lower monthly household incomes and rural residency are risk factors for child neglect. Juvenile delinquents whose major caregivers are “parents and grandparents” or “parents” may experience more child neglect. Therefore, caregivers should pay close attention to accompanying children and parent-child communication, rather than just ensuring material guarantee. The Chinese government should take necessary measures for the prevention and intervention of child neglect.

## Figures and Tables

**Figure 1 ijerph-20-04659-f001:**
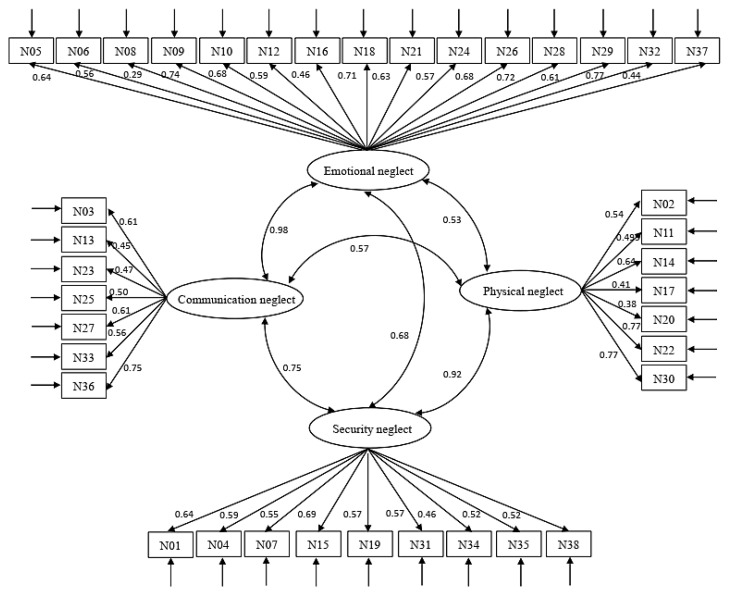
Model testing of the confirmatory factor analysis.

**Table 1 ijerph-20-04659-t001:** Demographic characteristics of the participants.

Variables	Categories	*n*	Percentage (%)
Childhood residence	Rural areas	124	58.5
	Suburbs	14	6.6
	City	69	32.6
BMI	<18.5	29	13.7
	18.5–22.9	133	62.7
	23–29.9	46	21.7
	≥30	3	1.4
Monthly household income	≤3000 RMB	25	11.8
	3001–5000 RMB	73	34.4
	5001–10,000 RMB	64	30.2
	≥10,000 RMB	46	21.7
Parents’ marital status	Original	147	69.3
	Single parent	35	16.5
	Remarried	28	13.2
Education level	Below junior high	17	8.0
	Junior high school	139	65.6
	Above junior high	55	25.9

**Table 2 ijerph-20-04659-t002:** Introduction of reverse scoring questions in the CNS.

Types	Items	Total
Emotional neglect	5, 6, **8**, 9, 10, 12, **16**, 18, 21, 24, 26, 28, 29, 32, **37**	15
Physical neglect	**2**, 11, **14**, 17, 20, **22**, **30**	7
Security neglect	**1**, **4**, **7**, **15**, 19, **31**, 34, 35, 38	9
Communication neglect	3, 13, 23, 25, 27, 33, 36	7

Note: bold ones are reverse scoring items.

**Table 3 ijerph-20-04659-t003:** CNS reliability (*n* = 212).

Dimension	Cronbach’s Alpha	Mean Inter-Item Correlations
Security neglect	0.813	0.325
Emotional neglect	0.897	0.370
Physical neglect	0.772	0.338
Communication neglect	0.753	0.310
Total neglect	0.939	0.294

**Table 4 ijerph-20-04659-t004:** Correlations among the total CNS and its four facets.

	1	2	3	4	5
1. Security neglect	——				
2. Emotional neglect	0.640 **	——			
3. Physical neglect	0.762 **	0.554 **	——		
4. Communication neglect	0.580 **	0.811 **	0.474 **	——	
5. Total neglect	0.849 **	0.926 **	0.769 **	0.848 **	——

Note: ** *p* < 0.01.

**Table 5 ijerph-20-04659-t005:** Confirmatory factor analyses (*n* = 212).

Model	χ^2^/df	GFI	IFI	TLI	CFI	RMSEA
Security neglect	1.888	0.954	0.958	0.936	0.957	0.065
Emotional neglect	2.485	0.883	0.905	0.883	0.904	0.084
Physical neglect	1.829	0.975	0.979	0.958	0.978	0.063
Communication neglect	2.178	0.962	0.946	0.916	0.944	0.075
Four-factor model	2.438	0.698	0.749	0.727	0.746	0.083

Note: GFI = Goodness of Fit Index, IFI = Incremental Fit Index, TLI = Tucker Lewis Index, CFI = Comparative Fit Index, and RMSEA = Root Mean Square Error of Approximation.

**Table 6 ijerph-20-04659-t006:** The descriptive statistics and *t*-test results of child neglect among juvenile delinquents.

Variable	Mean ± SD	Rural	Non-Rural	*t*	No Siblings	Have Siblings	*t*
		M ± SD	M ± SD		M ± SD	M ± SD	
SN	2.11 ± 0.72	2.20 ± 0.72	1.99 ± 0.69	−2.11 *	2.05 ± 0.57	2.14 ± 0.75	−0.876
EN	2.17 ± 0.73	2.24 ± 0.74	2.07 ± 0.70	−1.68	2.21 ± 0.76	2.14 ± 0.71	−0.605
PN	1.78 ± 0.68	1.87 ± 0.72	1.67 ± 0.62	−2.04 *	1.73 ± 0.67	1.81 ± 0.91	−0.809
CN	2.20 ± 0.75	2.31 ± 0.77	2.04 ± 0.69	−2.58 *	2.17 ± 0.70	2.21 ± 0.74	−0.403
TN	2.09 ± 0.63	2.18 ± 0.63	1.97 ± 0.59	−2.34 *	2.10 ± 0.63	2.10 ± 0.62	−0.213

Note: SN = Security neglect, EN = Emotional neglect, PN = Physical neglect, CN = Communication neglect, and TN = Total neglect, * *p* < 0.05.

**Table 7 ijerph-20-04659-t007:** One-way ANOVA results of child neglect among juvenile delinquents.

Variable	*n*	Mean ± SD				
		SN	EN	PN	CN	TN
**Age**						
15–17	125	2.13 ± 0.76	2.18 ± 0.74	1.82 ± 0.72	2.24 ± 0.80	2.11 ± 0.66
18–20	77	2.12 ± 0.69	2.19 ± 0.72	1.76 ± 0.63	2.17 ± 0.68	2.09 ± 0.58
21+	8	1.75 ± 0.25	1.75 ± 0.68	1.41 ± 0.35	1.86 ± 0.49	1.71 ± 0.32
*F*		/	1.37	1.39	1.04	1.60
**Education level**						
Below junior high	17	2.62 ± 0.87	2.45 ± 0.64	2.33 ± 0.66	2.40 ± 0.66	2.46 ± 0.52
Junior high school	139	2.08 ± 0.70	2.13 ± 0.74	1.77 ± 0.68	2.22 ± 0.77	2.07 ± 0.65
Above junior high	55	2.02 ± 0.64	2.16 ± 0.72	1.60 ± 0.56	2.06 ± 0.72	2.01 ± 0.56
*F*		5.07 **	1.48	8.30 **	1.60	3.66 *
LSD		1 > 2, 1 > 3	/	1 > 2, 1 > 3	/	1 > 2, 1 > 3
**Monthly household income**						
MHI1	25	2.21 ± 0.70	2.28 ± 0.79	2.13 ± 0.87	2.22 ± 0.73	2.23 ± 0.64
MHI2	73	2.10 ± 0.74	2.28 ± 0.79	1.82 ± 0.69	2.33 ± 0.79	2.16 ± 0.67
MHI3	64	2.25 ± 0.73	2.20 ± 0.68	1.79 ± 0.66	2.23 ± 0.76	2.14 ± 0.62
MHI4	46	1.90 ± 0.68	1.89 ± 0.61	1.57 ± 0.52	1.94 ± 0.64	1.84 ± 0.52
*F*		2.28	3.16 *	/	2.61	3.34 *
LSD		/	2 > 4	/	/	2 > 4, 3 > 4
**Major caregivers**						
Mother	36	2.11 ± 0.66	2.13 ± 0.68	1.57 ± 0.47	2.31 ± 0.78	2.06 ± 0.56
Father	9	2.09 ± 0.71	2.44 ± 0.75	1.91 ± 0.49	2.32 ± 0.90	2.24 ± 0.62
Grandparents	82	2.35 ± 0.75	2.42 ± 0.80	2.04 ± 0.73	2.43 ± 0.78	2.33 ± 0.66
Parents	59	1.95 ± 0.65	1.98 ± 0.63	1.67 ± 0.61	1.96 ± 0.63	1.91 ± 0.54
Parents & grandparents	22	1.61 ± 0.57	1.75 ± 0.46	1.32 ± 0.60	1.79 ± 0.49	1.65 ± 0.39
Others	4	2.58 ± 0.69	1.80 ± 0.30	2.29 ± 0.90	1.76 ± 0.54	2.07 ± 0.39
*F*		5.36 **	/	6.67 **	5.07 **	/
LSD		1 > 5, 3 > 4, 3 > 5, 5 < 6	/	1 < 3, 1 < 5, 2 > 5, 6 > 5, 3 > 4 > 5	1 > 4, 1 > 5, 3 > 4, 3 > 5	/

Note: MHI1 = Monthly household income ≤ 3000 RMB, MHI2 = 3001 ≤ Monthly household income ≤ 5000 RMB, MHI3 = 5001 ≤ Monthly household income ≤ 10,000 RMB, MHI4 = Monthly household income ≥ 10,000 RMB; SN = Security neglect, EN = Emotional neglect, PN = Physical neglect, CN = Communication neglect, and TN = Total neglect. LSD = Least significance difference, one of the post-hoc comparison methods of ANOVA. * *p* < 0.05, ** *p* < 0.01.

## Data Availability

Data will be made available on request.

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
