# Peer review of "Psychometric Properties of the Child Neglect Scale and Risk Factors for Child Neglect in Chinese Young Males Who Were Incarcerated"

_ijerph, 2023, doi:10.3390/ijerph20054659_

Round 1
Reviewer 1 Report
The topic of study fits well with the journal's scope and is a highly relevant, scientific, and social theme. Overall, the manuscript is presented in an intelligible way and is written in standard English. However, some topics need further clarification. Suggestions to address these concerns and improve the manuscript are outlined below.
Abstract
- It is uncommon to include statistical information in the abstract. I suggest removing such information from the abstract.
- The authors state that the "Child Neglect Scale may not have desirable psychometric properties with a four-factor model" but still analyze the different subscales. This seems nonsense.
Introduction
- Introduction needs some literature actualization. A considerable part of the literature has more than 5 years.
- There is an absence of a link between the first and the second paragraph of page 3. The introduction should have been clear if you used headings and subheadings. A clear and more robust link between the paragraphs should be included.
- One of the study's main goals is to conduct psychometric testing of the Child Neglect Scale, but you did not formulate any hypothesis regarding this specific goal. Moreover, the paper's title only reflects a part of the study. As you stated that one of your goals is to "delineate characteristics of child neglect in the Chinese cultural context" this goal should also be reflected in the title and abstract, as the abstract only refers to the goal of examining the psychometric properties of the CNS and not the second objective, but results regarding the second objective are presented. The introduction would benefit from a reformulation considering the study's two main objectives.
Procedures
- The authors referred that: “participants were randomly recruited from two districts of the juvenile detention center”. How was the randomization performed? Information on randomization should be included.
Measures
- How information regarding participants' residence, BMI, income, parents' marital status, and education level were collected? The measures section should include information on how sociodemographic data was gathered.
Results
- Why did you call Parallel validity instead of convergent validity?
- The four-factor model’s fit indices were undesirable; however, you decided to present the results based on the four factors (subscales (i.e., the characteristics of child neglect section). Why? What are the main limitations of such a decision? How reliable are your results?
- Effect size data should be computed (both for t-tests and ANOVAs).
Discussion
- Again, you only present the objective of examining the psychological properties of the CNS. You should be more precise when you present your study's objectives (and results), as it seems that your study is not only focused on CNS psychometric properties.
- In the discussion, you refer: "Therefore, the CNS may not be appropriate to measure child neglect based on a four-factor model, but separate subscales for measuring child neglect in juvenile delinquents in the Chinese cultural context”. This conclusion is very confusing for me. How can you say that the four-factor structure is not appropriate to measure child neglect, but separate subscales are appropriate for measuring child neglect? Especially taking into account what you say in lines 364-366 (i.e., CNS may not accurately and truly show the characteristics of child neglect of juvenile delinquents in this study).
- Data regarding the other tested models that you include in the discussion section (page 10, lines 377-380), in my opinion, should be briefly described in the results section.
- You refer: “Moreover, some items that belong to the same subscale are very similar in content. These may cause confusion and fatigue among the participants and then affect the validity of the CNS”. Did you check for item loadings in each factor? Were there loadings lower than 0.3? Did you consider removing these items?
Minor comments
Some expressions should be avoided in a written work, such as “didn’t”, “can’t”.
Author Response
Dear Ms. Tamara Ovari,
Thank you very much for giving us an opportunity to resubmit a revised manuscript. We also appreciate editors and reviewers very much for their constructive comments and suggestions on the manuscript entitled “Psychometric properties of the Child Neglect Scale in Chinese young males who were incarcerated” (Manuscript Number: ijerph-2198781). We have studied reviewers’ comments and we have tried our best to revise the manuscript according to the comments, and the revised version has been submitted into the website. Please find the detailed information in the revised manuscript. Thank you and best regard.
List of ResponsesDear Editors and Reviewers, Thank you for your letter and for the reviewers’ comments concerning our manuscript entitled “Psychometric properties of the Child Neglect Scale in Chinese young males who were incarcerated”(Manuscript Number: ijerph-2198781). Those comments are all valuable and very helpful for revising and improving the manuscript. We have studied the comments carefully and have made corrections which hope meet with approval. The main corrections in the manuscript and the responses to the reviewers’ comments are following:
Responses to the Reviewers’ comments:
#REVIEWER 1
The topic of study fits well with the journal's scope and is a highly relevant, scientific, and social theme. Overall, the manuscript is presented in an intelligible way and is written in standard English. However, some topics need further clarification. Suggestions to address these concerns and improve the manuscript are outlined below.
Abstract
Comment 1- It is uncommon to include statistical information in the abstract. I suggest removing such information from the abstract.
Response: Thank you very much for your suggestion, we have removed the information from the Abstract.
Comment 2- The authors state that the "Child Neglect Scale may not have desirable psychometric properties with a four-factor model" but still analyze the different subscales. This seems nonsense.
Response: Thank you very much for your comment, we have revised this issue in the Abstract.
Introduction
Comment 3- Introduction needs some literature actualization. A considerable part of the literature has more than 5 years.
Response: Thank you very much for your comment. We have revised this issue in the manuscript, please find the detailed information in the revised manuscript. Moreover, 50.82% of citations are references in recent five years (2017-2022).
Comment 4- There is an absence of a link between the first and the second paragraph of page 3. The introduction should have been clear if you used headings and subheadings. A clear and more robust link between the paragraphs should be included.
Response: Thank you so much for your comment. We have revised it in the manuscript, please find the detailed information in the revised manuscript.
Comment 5- One of the study's main goals is to conduct psychometric testing of the Child Neglect Scale, but you did not formulate any hypothesis regarding this specific goal. Moreover, the paper's title only reflects a part of the study. As you stated that one of your goals is to "delineate characteristics of child neglect in the Chinese cultural context" this goal should also be reflected in the title and abstract, as the abstract only refers to the goal of examining the psychometric properties of the CNS and not the second objective, but results regarding the second objective are presented. The introduction would benefit from a reformulation considering the study's two main objectives.
Response: Thank you so much for your comment. We have revised it in the manuscript, please find the detailed information in the revised manuscript.
Procedures
Comment 6- The authors referred that: “participants were randomly recruited from two districts of the juvenile detention center”. How was the randomization performed? Information on randomization should be included.
Response: Thank you very much for your comment. We have revised this issue in the manuscript, please find the detailed information in the revised manuscript, such as “Specifically, the authors randomly chose room numbers, and prison guards asked the participants who lived in these rooms to finish the questionnaires.”.
Measures
Comment 7- How information regarding participants' residence, BMI, income, parents' marital status, and education level were collected? The measures section should include information on how sociodemographic data was gathered.
Response: Thank you very much for your comment. We have revised this issue in the manuscript, please find the detailed information in the revised manuscript, such as “Demographic Information Questionnaire (DIQ). The DIQ was developed by the authors, which used to collect demographic information about the participants, including height, weight, family income, parents’ marital status, and educational levels.”.
Results
Comment 8- Why did you call Parallel validity instead of convergent validity?
Response: Thank you very much for your comment. We have revised this issue in the manuscript, please find the detailed information in the revised manuscript.
Comment 9- The four-factor model’s fit indices were undesirable; however, you decided to present the results based on the four factors (subscales (i.e., the characteristics of child neglect section). Why? What are the main limitations of such a decision? How reliable are your results?
Response: Thank you very much for your comment. The original version of the CNS based on normal young adults is a four-factor model, and the current study want to replicate the four-factor model in youth who were incarcerated. However, the four-factor model of the CNS had undesirable model fit. Moreover, the model fits for these four independent subscales were acceptable, which suggested that the CNS had four scales to examine different kinds of neglect, respectively.
Comment 10- Effect size data should be computed (both for t-tests and ANOVAs).
Response: Thank you very much for your comment. We have revised this issue in the manuscript, please find the detailed information in the revised manuscript.
Discussion
Comment 11- Again, you only present the objective of examining the psychological properties of the CNS. You should be more precise when you present your study's objectives (and results), as it seems that your study is not only focused on CNS psychometric properties.
Response: Thank you very much for your comment. We have revised this issue in the manuscript, please find the detailed information in the revised manuscript.
Comment 12- In the discussion, you refer: "Therefore, the CNS may not be appropriate to measure child neglect based on a four-factor model, but separate subscales for measuring child neglect in juvenile delinquents in the Chinese cultural context”. This conclusion is very confusing for me. How can you say that the four-factor structure is not appropriate to measure child neglect, but separate subscales are appropriate for measuring child neglect? Especially taking into account what you say in lines 364-366 (i.e., CNS may not accurately and truly show the characteristics of child neglect of juvenile delinquents in this study).
Response: Thank you very much for your comment. We have revised this issue in the manuscript, please find the detailed information in the revised manuscript.
Comment 13- Data regarding the other tested models that you include in the discussion section (page 10, lines 377-380), in my opinion, should be briefly described in the results section.
Response: Thank you very much for your comment. We have revised this issue in the manuscript, please find the detailed information in the revised manuscript.
Comment 14- You refer: “Moreover, some items that belong to the same subscale are very similar in content. These may cause confusion and fatigue among the participants and then affect the validity of the CNS”. Did you check for item loadings in each factor? Were there loadings lower than 0.3? Did you consider removing these items?
Response: Thank you very much for your comment. We have revised this issue in the manuscript, please find the detailed information in the revised manuscript.
Minor comments
Comment 15-Some expressions should be avoided in a written work, such as “didn’t”, “can’t”.
Response: Thank you very much for your comment. We have revised this issue in the manuscript, please find the detailed information in the revised manuscript.
Reviewer 2 Report
Dear authors, thank you for giving me the opportunity to review your manuscript: “Psychometric properties of the Child Neglect Scale in Chinese young males who were incarcerated”.
The article is well written, pertinent, and can meaningfully contribute to the literature.
I send below some comments to improve the manuscript.
Abstract:
· Authors described: Child neglect is an important risk factor for juvenile delinquency, while few studies have examined child neglect in juvenile delinquents due to the lack of appropriate measurement tools.
This sentence is incorrect since several studies have examined child neglect in juvenile delinquents in other countries. Please add Chinese reality to the sentence (for example: …few studies have examined child neglect in Chinese juvenile delinquents…)
Introduction:
· Authors described: Existing studies have suggested that plenty of factors increase the likelihood of juvenile delinquency, especially adverse childhood experiences (e.g., Piquero et al., 2003; Ryan et al., 2013). – The references they use are old. Please update some references throughout the introduction and discussion. There is, for example, a recent systematic literature review on this topic [Pires, A.R. & Almeida, T.C. (2023). Risk Factors of Poly-victimization and the Impact on Delinquency in Youth: A systematic review. Crime & Delinquency. https://doi.org/10.1177/00111287221148656].
Discussion:
· Please provide more recent references throughout the discussion.

Author Response
Dear Ms. Tamara Ovari,
Thank you very much for giving us an opportunity to resubmit a revised manuscript. We also appreciate editors and reviewers very much for their constructive comments and suggestions on the manuscript entitled “Psychometric properties of the Child Neglect Scale in Chinese young males who were incarcerated” (Manuscript Number: ijerph-2198781). We have studied reviewers’ comments and we have tried our best to revise the manuscript according to the comments, and the revised version has been submitted into the website. Please find the detailed information in the revised manuscript. Thank you and best regard.
List of ResponsesDear Editors and Reviewers, Thank you for your letter and for the reviewers’ comments concerning our manuscript entitled “Psychometric properties of the Child Neglect Scale in Chinese young males who were incarcerated”(Manuscript Number: ijerph-2198781). Those comments are all valuable and very helpful for revising and improving the manuscript. We have studied the comments carefully and have made corrections which hope meet with approval. The main corrections in the manuscript and the responses to the reviewers’ comments are following:
Responses to the Reviewers’ comments:
#REVIEWER 2
Dear authors, thank you for giving me the opportunity to review your manuscript: “Psychometric properties of the Child Neglect Scale in Chinese young males who were incarcerated”. The article is well written, pertinent, and can meaningfully contribute to the literature. I send below some comments to improve the manuscript.
Abstract:
Comment 1-Authors described: Child neglect is an important risk factor for juvenile delinquency, while few studies have examined child neglect in juvenile delinquents due to the lack of appropriate measurement tools.
This sentence is incorrect since several studies have examined child neglect in juvenile delinquents in other countries. Please add Chinese reality to the sentence (for example: …few studies have examined child neglect in Chinese juvenile delinquents…)
Response: Thank you very much for your comment. We have revised this issue in the manuscript, please find the detailed information in the revised manuscript.
Introduction:
Comment 2-Authors described: Existing studies have suggested that plenty of factors increase the likelihood of juvenile delinquency, especially adverse childhood experiences (e.g., Piquero et al., 2003; Ryan et al., 2013). – The references they use are old. Please update some references throughout the introduction and discussion. There is, for example, a recent systematic literature review on this topic [Pires, A.R. & Almeida, T.C. (2023). Risk Factors of Poly-victimization and the Impact on Delinquency in Youth: A systematic review. Crime & Delinquency. https://doi.org/10.1177/00111287221148656].
Response: Thank you very much for your comment. We have revised this issue in the manuscript, please find the detailed information in the revised manuscript.
Discussion:
Comment 3-Please provide more recent references throughout the discussion.
Response: Thank you very much for your comment. We have revised this issue in the manuscript, please find the detailed information in the revised manuscript. Moreover, 50.82% of citations are references in recent five years (2017-2022).